# Sustainable Management Practices for Urban Green Spaces to Support Green Infrastructure: An Italian Case Study

**Monica Pantaloni** [1,*], **Giovanni Marinelli** [2], **Rodolfo Santilocchi** [1], **Alberto Minelli** [3] **and Davide Neri** [1]

1   Department of Agricultural, Food, and Environmental Sciences, Polytechnic University of Marche, 60131 Ancona, Italy; r.santilocchi@staff.univpm.it (R.S.); d.neri@staff.univpm.it (D.N.)
2   Department of Materials, Environmental Sciences and Urban Planning, Polytechnic University of Marche, 60131 Ancona, Italy; g.marinelli@staff.univpm.it
3   Department of Agricultural and Food Sciences, Alma Mater Studiorum, University of Bologna, 40127 Bologna, Italy; alberto.minelli@unibo.it
*   Correspondence: m.pantaloni@staff.univpm.it; Tel.: +39-071-2204695 or +39-339-1491765

**Abstract:** Traditional land-use planning models have proven inadequate to address contemporary issues in sustainable development and protection governance. In recent years, new 'performance based' approaches that integrate ecosystem services (ES) provided via green infrastructure (GI) into traditional spatial planning models have been proven to reach a higher level of environmental performance, necessary to improve quality of life for all people. In Italy, there are no mandatory planning instruments to design and manage GI, which still remains a component of the traditional land-use plan. Here, the development of urban green spaces (UGS) based on 'quantitative assessment' is not suitable for guaranteeing the supply of ES. In addition, the scarcity of financial resources to develop 'green standards', as prescribed in the land-use plan to strategically design the GI, is an issue for most Italian public administrations. The paper provides the results of a test case conducted in a public green area of the city of Ancona, where the experimentation of a diversified maintenance strategy of an urban lawn significantly reduced the management cost and improved the environmental performance of green spaces. The identification of a unified management strategy to be applied to all the public UGS can help to achieve better results in support of sustainability, to redesign the continuity of GI and to develop strategies for future urban green master plans.

**Keywords:** green infrastructure; ecosystem services; green spaces management; environmental performance

## 1. Introduction

Since 1987, with the publication of Our Common Future [1] which defined the political concept of sustainability for the first time, international debate has been aimed at consolidating a new growth model that aims to reconcile the two aspects of sustainable growth, integrating environmental concerns into decision making [2,3].

To transfer these concepts into a spatial planning approach, it became necessary to reconsider the traditional land-use planning models (based exclusively on functional zoning, i.e., residential, manufacturing, commercial), which have proven inadequate to address contemporary issues in sustainable development and protection governance [4].

To do this, in recent years, new approaches based on the integration of ecosystem services (ES) [5,6] have been considered as an alternative to traditional spatial planning models, to reach the higher level of environmental performance that is necessary to improve the quality of life of all people and preserve resources for future generations [7].

In these studies, the provision of ES via spatial planning [8,9] is guaranteed by the adoption of green infrastructure (GI), defined as 'a strategically planned network of natural and semi-natural areas with other environmental features designed and managed to deliver

a wide range of ecosystem services' [10], such as supporting, provisioning, regulating and cultural services [11].

At the city scale, GI is considered a conceptual tool to plan, design and manage a framework of green spaces [12], which represent key elements for the conservation and management of biodiversity and landscapes [13].

Despite awareness of shifting the focus away from traditional land-use planning to a performance based approach supported by the integration of ES and GI, the development of public green spaces in the urban context is already based on achieving the 'quantities of green' (that links, for example, green spaces availability to urban surfaces, or per capita public green areas availability) that are considered the most representative sustainable indicators to guarantee the 'maintenance of natural capital' [14–17].

On the contrary, a 'qualitative' assessment of urban green spaces (UGS) based on the estimation of different levels of provision (i.e., UGS availability, accessibility, attractiveness, and usability) will better express the real contribution of UGS in the improvement of the environmental, social, and economic performance of GI [18]. Furthermore, the management of green space, supported by an adequate organizational strategy, is fundamental to providing ecosystem services at the municipal level [19].

In Italy, there are no mandatory planning instruments to design and manage the GI, which still remains a component of the traditional land-use plan.

At the spatial scale, a land-use plan transposes the environmental value derived from mandatory regional and provincial planning instruments (i.e., landscape and territorial plans) or nonmandatory sectoral instruments (i.e., the ecological network).

At the city scale, a land-use plan identifies the existing public and private UGS and also provides for the development of new urban and peri-urban green areas but only based on quantitative indicators (i.e., the achievement of a 'minimum quantity of greenery per capita') set by planning standards.

This 'quantitative' evaluation approach, imposed by the Ministerial Decree of the Italian urban planning regulation, is not suitable for guaranteeing a high level of environmental performance based on ES [4].

In addition, the Italian municipalities responsible for the design and the management of UGS can hardly find the financial resources to build a new UGS, defined as 'green standards', that help deliver ES via GI.

Recently, in 2020 and with the approval of the Italian Ministerial Decree for the public green spaces [20], a set of instruments has been recognized—such as a (a) green census, (b) green master plan, (c) green management plan, and (d) green regulation—to support local decision makers to manage urban and peri-urban green areas. In particular, the green master plan represents the most important long term strategic tool aimed at planning the interventions to design and manage GI, also considering the economic resources to do so. This set of instruments promotes and facilitates the planning and design of GI and ES at the city scale, focusing on the integration between the public UGS (existing and new ones) recognized by the land-use plan and the management aspect of UGS.

With this approach, combining technical solutions within the planning tool has become a priority to find financial resources for the sustainable management of ES via GI, taking into account the maintenance costs related to their provision.

For this reason, supporting sustainable strategies in the management of public UGS would allow the development of the 'green standards' set out in the land-use plan, with progressive growth in the GI.

To achieve this, the application of the diversified maintenance and management of the urban lawn, which is the main component of UGS [21], may be one possible approach to reduce the management costs of urban greenery. Consequently, the evaluation of its effectiveness at the city scale is needed.

In this study, we aim to demonstrate that a management strategy based on qualitative standards can enhance green performance at the city scale, providing ecosystem services with a significative costs reduction.

For this purpose, we firstly provide a method based on 'qualitative indicators' to classify all the green areas of the Municipality of Ancona and define GI at the urban scale.

Secondly, we analyze the application of the diversified maintenance and management of the urban lawn at the local scale, by presenting a case study conducted in a public green area of the city of Ancona. Specifically, we present and discuss the main results obtained by the application of a diversified maintenance approach in turf management on the campus of the Marche Polytechnic University.

Finally, we discuss how the results obtained at the local scale can bring implications in UGS at the city scale, by outlining a future scenario for the GI.

## 2. Materials and Methods

### 2.1. Evaluation of Green Infrastructure at City Scale

The city of Ancona (43°37′ N 13°31′ E) is the primary city and regional capital of the Marche region (central Italy) with a seaport located in the middle of the Italian Adriatic coast. The municipal area of Ancona is the region's largest city. The total land area measures 124.84 km$^2$, with a total population of 100,497 and a population density of 805 (inhabitants/km$^2$) [22].

This part of the study offers preliminary evaluations of the total amount of urban greenery and classifies different types of green spaces (as a framework of the green infrastructure of the municipality of Ancona) through the application of both quantitative and qualitative indicators [23–25]. They help to evaluate the ratio with the total impermeable surfaces, the level of fragmentation of urban greenery, spatial distribution, and the relationship with other public spaces, allowing us to define size and shape of existing GI [26].

To assess the quantity of GI (considering just the public areas that include vegetation elements), three indicators were applied: (a) percentage of green on the municipal surface (excluding natural protected areas); (b) per capita green areas availability (m$^2$/inhabitant); and (c) total percentage of green.

Quantitative indicators have been taken from the Italian Institute for Environmental Protection and Research, ISPRA elaborations, based on the Statistics National Institute (ISTAT) for the 109 municipalities that are provincial capitals, and ISPRA/SNPA (the national network of environmental protection agencies) for the 15 among the most populated Italian municipalities.

For the city of Ancona, value corresponding to point (c) is higher than in other provincial capitals due to the inclusion of the natural area of the Conero Regional Park (considered as a natural reserve in urban area) [27] (Table 1).

**Table 1.** Percentage of public green areas on the municipal surface, availability per capita and total percentage of green (year 2018, a2017). Marche region—center of Italy.

| Municipality/Provincial Capital | Percentage of Green on the Municipal Surface (%) | Per Capita Green Areas Availability (m$^2$/Inh) | Percentage of Total Green (Public Green Areas + Natural Protected Areas, Net to Overlapping) |
|---|---|---|---|
| Ancona | 4.2 | 51.3 | 29.2 |
| Ascoli Piceno | 0.3 | 9.4 | 9.8 |
| Macerata | 0.9 | 20.0 | 0.9 |
| Pesaro | 2.1 | 27.9 | 22.6 |

Analyzing geospatial information of the land-use plan, 'public green spaces' for residential zones, defined as 'standard' in the land-use category (according to National Decree 1444/68), accounts for 3.82 km$^2$.

This data exceeds the minimum value of 9 m$^2$/inhabitant imposed by the Ministerial Decree (which corresponds to 902.538 m$^2$, based on the size of the population).

The information on the total amount of the urban green standards (not in the public domain) was obtained through direct communication with the officials of the offices of the Municipality of Ancona.

To assess the quality of GI [28], we analyzed geo-databases considering just the public part of the green surface, excluding private green spaces.

Green surfaces derived the land-use plan reached from satellite imagery and processed in a GIS environment to allow for spatial distribution and visualization of GI.

We classified the total green area by applying the univocal attribution of the following 13 typologies of urban green areas, identified by an interinstitutional research group (Ispra, Istat and the Italian Ministry of the Environment, Land and Sea):

1.  Historical green—L.D. 42/2004;
2.  Urban parks (area > 8000 m$^2$);
3.  Equipped green: parks and neighborhood gardens (area < 8000 m$^2$);
4.  Decorative green areas;
5.  Urban forestry;
6.  School gardens;
7.  Botanical gardens;
8.  Urban allotments (vegetable garden);
9.  Outdoor sports areas;
10. Wood areas (area > 5000 m$^2$);
11. Uncultivated green (suburban park, geological and botanical emergencies); and
12. Cemeteries.

Categories are based on differentiation in structure (dimensions, species composition, presence of water . . . ) and the ecological and social-cultural function of the ecosystem services [29].

Lastly, we introduced 'usability' and 'management intensity' as indicators to qualitatively differentiate UGS at city scale. A valid definition of 'usability' has been taken from the Marche Regional Law n. 63/2015 and the literature, as follows:

"Usability of green spaces is defined as the possibility and degree of use by the citizens of a specific green area. [ . . . ]" also measured by the presence of facilities able to support people's activity [30] which determines level of equipment.

Instead, according to Young and Delshammar [19,31], we define urban green space 'management' as:

"[ . . . ] a range of activities that includes different aspect of sustainable urban development (social, ecological, economic) and its role in providing ecosystem services with a strategic, user-orientated approach that considers not just the 'maintenance' of vegetation but the relationship between open spaces and users and the activities related to these".

This definition helps to classify the green areas with a predominantly recreational and social function [32,33], mainly related to cultural services, i.e., recreation and tourism [11].

The indicators are assessed (Table 2) and ranked individually by assigning a value between 1 and 5, according to the criteria described for each level (1 low performance; 2 medium/moderate; 5 high performance).

**Table 2.** Indicators, evaluation criteria and level of estimation to classify green areas by qualitative approach.

| Indicator | Established Value | Characteristics |
|---|---|---|
| Usability | 1 | Green surfaces that present low levels of accessibility and equipment; mainly assigned to natural and protected areas. |
| | 2.5 | Areas with historical and aesthetic value or environmental protection, and a good level of accessibility. |
| | 5 | Neighborhood areas that mainly assume a recreational and social function and present a high level of accessibility. |
| Management | 1 | The main purpose is to preserve biodiversity and ecological services (seminatural area). We assigned this value to UGS that mainly provide provisioning plus regulation and maintenance services as dominant categories over others. |
| | 2.5 | UGS that present a balance between the provision of regulation services, but also cultural services such as aesthetic information, inspiration for culture, art and design, spiritual experiences. |
| | 5 | Areas with the highest level of usability for citizens, which is related to the highest intensity of use. UGS are characterized by the provision of cultural services such as intellectual and experiential ones. |

This rapid assessment methodology made it possible to divide the total number of UGS into three levels (Table 3) that link together the indicators of 'usability' and 'management' (Figure 1).

**Table 3.** Three levels according to threshold index. 'Uvalue' means the value assigned to usability indicator, 'Mvalue' means the value assigned to management indicator.

| Level | Threshold | Formula |
|---|---|---|
| 1 | <5 | |
| 2 | ≥5; ≤7.5 | Uvalue + Mvalue |
| 3 | >7.5 | |

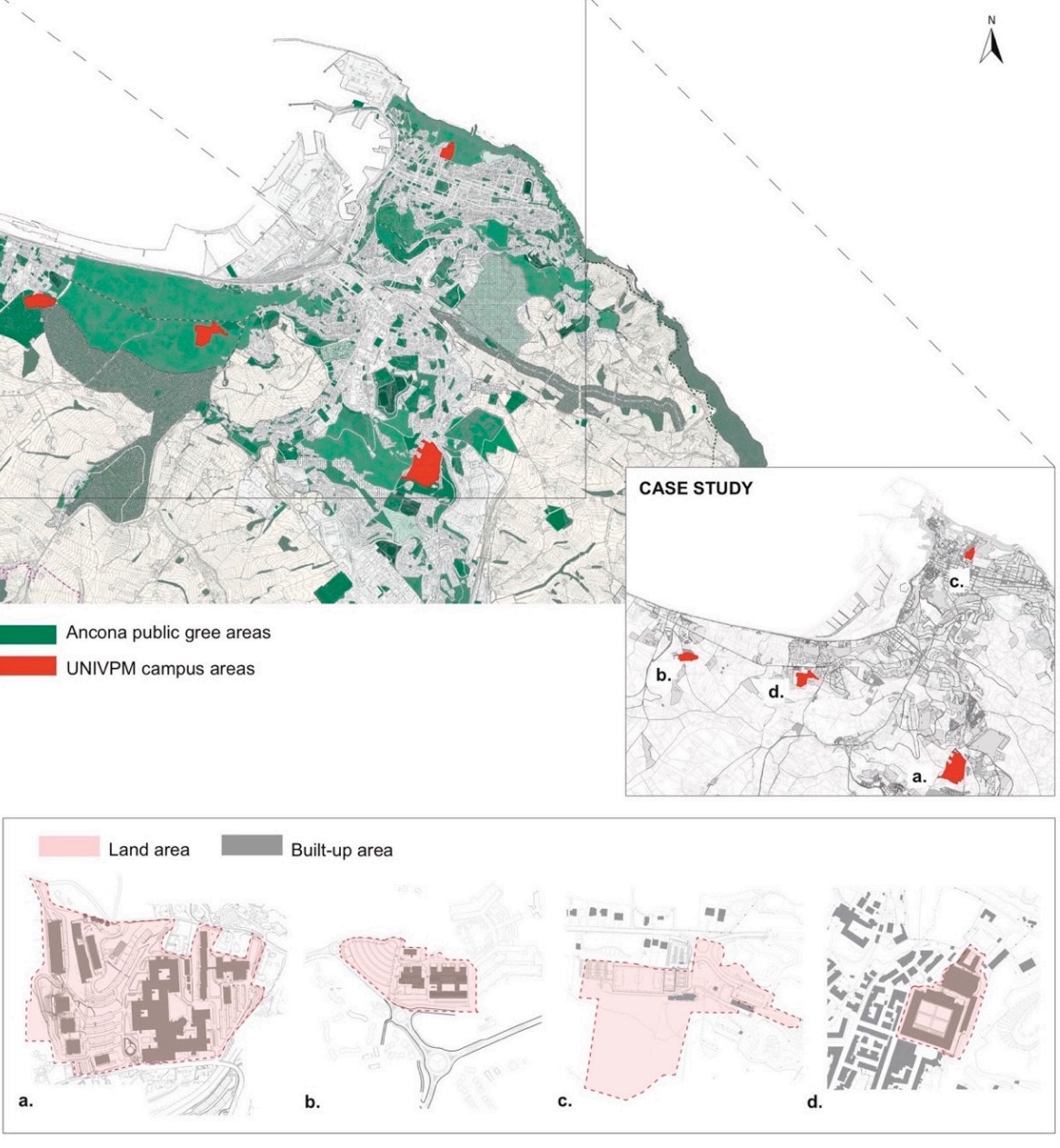

**Figure 1.** Existing UGS in Ancona with green areas of university (red). Source: author's own elaboration from data taken by the land-use plan, Municipality of Ancona, year 2018. (**a**) Montedago (**b**) Torrette (**c**) city center (**d**) Posatora.

*2.2. Case Study: Green Areas in Marche Polythecnic University Campus*

The experimental study was developed in the green areas of the university campus, which is divided into four sites located in Ancona's neighborhoods: (a) Montedago center, where Faculty of Engineering, Science and Agriculture are settled; (b) Faculty of Medicine in Torrette; (c) Faculty of Economics in the city center; and (d) university sports center, located in Posatora (Figure 1).

As we performed for the GI at city scale, we carried out quantitative and qualitative evaluation starting from a quantitative assessment of the total of green areas, by means of a visual census and geometric surveys.

Starting from the total land area of the campus, uncovered/external area was evaluated, which counted for about 78% of the total land area. A total of 45% of this are pavement areas, while the percentage of total green areas was quantified by dividing built up areas and pavement areas by the total land area. This corresponds to about 43% of the total land area of the university campus (Table 4).

**Table 4.** Quantification of university campus green areas.

| University Campus Faculties-Departments | City Location | Land Area (m²) | Built Up Area (m²) | University Campus Total External Areas (m²) | Pavement Areas (m²) [1] | Total Green Areas (m²) |
|---|---|---|---|---|---|---|
| Engineering, Agriculture, Biology, | Montedago | 148,974 | 36,305 | 112,628 | 50,683 | 61,945 |
| Medicine | Torrette | 38,392 | 8488 | 29,902 | 22,555 | 7347 |
| Economics | City center | 22,096 | 8262 | 13,834 | 8405 | 5429 |
| Sports centre | Posatora | 53,526 | 3440 | 50,086 | 9562 | 40,524 |
| Total | | 262,988 | 56,495 | 206,450 | 91,205 | 115,245 |

[1] pedestrian, viability and parking areas.

Qualitative evaluation was carried out by differentiating type of vegetation and classifying them into four categories: trees, lawns, meadows, and cultivated green areas (Figure 2).

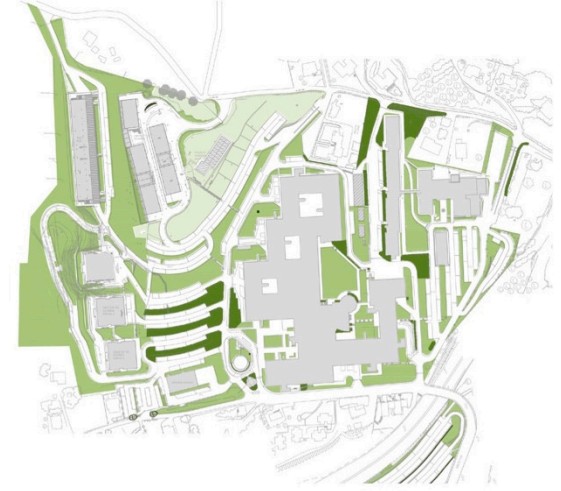

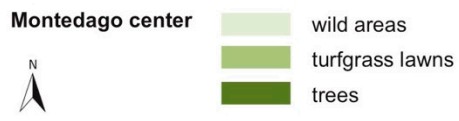

| City location | Trees (m²) | wild (m²) | cultivated areas (m²) | turfgrass lawns (m²) | total green areas (m²) |
|---|---|---|---|---|---|
| Montedago | 6172 | 9405 | 135 | 46232 | 61945 |
| Torrette | 724 | - | - | 6623 | 7347 |
| City center | 793 | - | - | 4636 | 5429 |
| Posatora | 8734 | 18520 | - | 13270 | 40524 |
| Total | 16,423 | 27,925 | 135 | 70,761 | 115,245 |

**Figure 2.** Classification by typology of campus green areas. Diagram (left side) refers to Montedago center, at the current situation.

Turfgrass lawns count for about 60% of the total university campus green areas, resulting in the main component of horizontal green in terms of size.

Green Management Strategy

Management of these areas is entrusted to the Polytechnic University of Marche, specifically to the Didactic and Experimental Farm 'Pasquale Rosati', which used to belong to the Department of Agricultural, Food and Environmental Science, D3A. During the experimentation period, change in ordinary maintenance practices based on a more sustainable approach in management of green space was evaluated.

In 2006 it was decided to shift the focus from an 'aesthetical' to a more 'functional' approach to green spaces design and management, considering their importance in providing benefits to humans and their role in whole landscape design at the urban scale.

In addition, this was due to shrinking financial availability and growth in the total land area of university campus, from 14 to 21 hectares, including green areas (about 40% increase of areas to be managed).

The observation started in 2006 and ended in 2019. Management policies was based on three principles:

1. Differentiation in the maintenance of turfgrass lawns;
2. Improvement of machinery equipment to eliminate pesticide and herbicide applications and implement alternative solutions to support urban biodiversity; and
3. Sustainable use of water based on the reduction of withdrawals.

Different management practices associated with the second and third criterion are specifically designed according to the differentiation of lawns (first principle).

The first principle brought to the categorization of the turf lawn by *location* and *typology*. This has been essential to identify the degree of intensity of benefits for the users concerning the costs, necessary for the development and maintenance of such area [34]. In fact, the use of the appropriate machinery, the height, and the frequency of the cut depends on the type of turf in relation to the different functionalities. Additionally, the height and frequency of mowing were performed in order to respect the "1/3 rule", which means that mowing does not involve more than one third of the total height of the grass, in order to prevent scalping and physiological stress [35–37]. These aspects interact with other factors, such as the frequency and quantity of watering and fertilization.

With a broader view to environmental sustainability, in fact, in the transitional areas, in countries facing the Mediterranean Sea, the selected and most used turfgrass species in recent years were warm season turfgrass species, with the aim of reducing the consumption of water, herbicides and, in any case, to obtain good quality turf [38].

In application of the first principle, the total UGS was divided into three types of areas that align with three levels of maintenance according to their functionality and ornamental value (Figure 3):

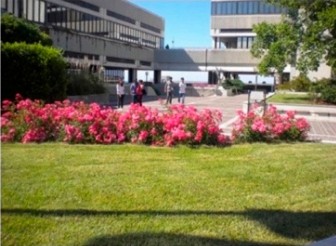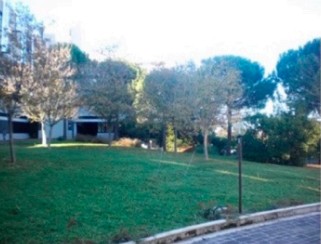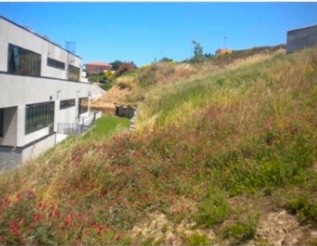

**Figure 3.** From left: examples of type 1, type 2, type 3 areas of the UNIVPM campus in Montedago center.

Type 1 identifies an intensive level of maintenance applied to the areas of great ornamental value, mainly located near the entrances and principal courtyards. In this type of area, the floral attraction is maintained according to precise annual plantings, their lawns are mowed frequently, and cleaning operations are performed, even, daily.

Type 2 is related to normal level maintenance, associated with a high level of usability for study activities, recreational purposes, and socialization. In this type of area, mow-

ing comes no more than ten times in the growing season, and cleaning is scheduled at planned intervals.

Type 3 corresponds to seminatural maintenance, with low intensity interventions in extensive peripheral areas not heavily frequented, where the purpose is to preserve biotic features of the ecosystem. In these types of areas, the grass is mowed no more than three times in the growing season, trees and shrubs are pruned only when necessary and cleaning is performed on a planned schedule.

Thus, from 2006 to 2019, routine maintenance operations were progressively reduced in each area type described above. Mechanization and water consumption depended on the diversified maintenance and biological life cycles during the seasons.

For type 1 areas, grass mowing was reduced from 1 turn/7 day to 1 turn/10 days from March to September, and from 1 turn/10 day to 1 turn/15 days in the winter months, with basket grass collection and leaf removal. Working days passed from 16 to 8 per year. No mulch mowing was used. Fertilization was reduced from 4 to 2 operations per year. Irrigation depended on seasons.

For type 2 areas, grass mowing was reduced from 1 turn/8–10 days to about 1 mowing per 2 months with basket grass collection, depending on season. In winter months maintenance operation were cut. Depending on frequency of use and aesthetic value of the green areas, instead of being mowed, the lawn was shredded without collection. This operation followed biological lifecycle during the seasons and aimed to encourage growth of spontaneous shrubs and weed. Routine maintenance such as fertilization and irrigation were cut.

For type 3 areas, shredding without collection was less intensive, aiming to support a greater abundance of diversity and spontaneous self-seeding of the herbaceous species. It was limited to once per year (May/June). It also aims to encourage the growth of spontaneous shrubs and weed control.

Shredding was scheduled to support self-seeding of annual and biennial spontaneous herbaceous plants and the growth of spontaneous bushes, and to limit the development of aggressive perennial plants, i.e., bramble. No fertilization or irrigation was carried out (Table 5).

**Table 5.** Comparisons of the maintenance operations applied at the beginning of the experimentation (year 2006) and at the end of the experimentation (year 2019).

| Area Type | Turf Management | | | | | Water Management | |
| --- | --- | --- | --- | --- | --- | --- | --- |
| | Grass Mowing | | Grass Shredding | Fertilization | | Irrigation | |
| | year 2006 | year 2019 | 2019 | 2006 | 2019 | 2006 | 2019 |
| (1) | One turn/seven days (March–September) One turn/ten days (winter months) basket grass collection + leaf removal | One turn/ten days (March–September) One turn/fifteen days (winter months) | - | Four/six times per year | Twice per year | June–September | June–September |
| (2) | One turn/eight–ten days basket grass collection | One turn per two months (no maintenance in winter month) | - Two shredding turns in spring time and autumn - muching | - | - | emergency | - |
| (3) | - | - | - Once per year (May/June) - muching | - | - | - | - |

The efficiency of maintenance operations increased thanks to the investment in machinery equipment with a useful life of 5 to 10 years. In 2019, investment of EUR 90,000 was 4 times higher than the total investment in 2006.

Improvement in machinery consisted of: (a) employment of n.5 grass trimmer, n.3 lawnmowers, n.1 chainsaw, n.1 hedge trimmers, n.1 brush cutter arm; (b) buying

new equipment consisting in n.1 mulcher, n.1 lawn mower, n.1 chipper, n.1 brushing-collecting machine. These have made it possible to avoid leaf removal and waste disposal from pruning, reusing organic material as fertilizer.

For cleaning streets, brushing-collecting machines allowed the management of weeds on the road verges and sidewalks by mechanical shredding.

For more sustainable use of water, water waste was avoided and clean water used for irrigation. Since 2006, irrigated lawns have been maintained in type 1 areas, decreasing the rotational irrigation. In 2019, for type 2 and 3 areas, lawns were designed to avoid mechanical irrigation.

## 3. Results

At city scale, through the differentiation of green areas by typology, we defined the spatial distribution and percentage values of different type of UGS (Figure 4).

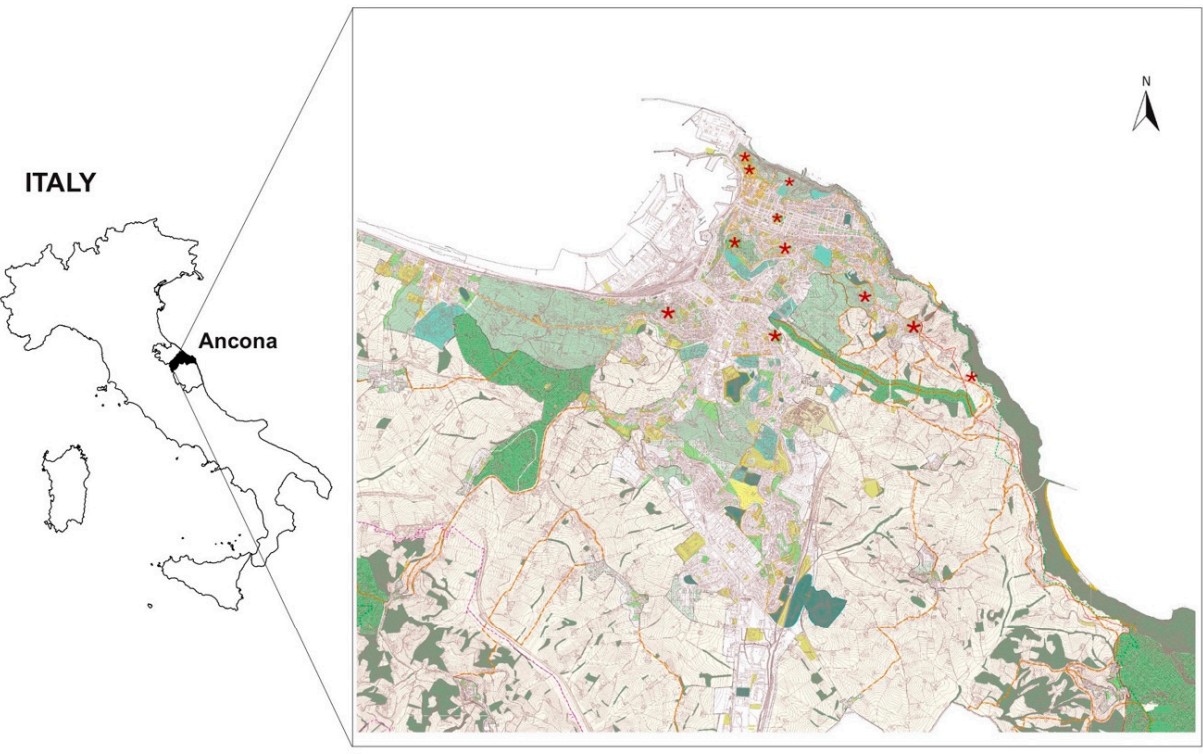

### percentage composition of public green typologies* divided in three level

| LEVEL 1 | | | | LEVEL 2 | | | | LEVEL 3 | |
|---|---|---|---|---|---|---|---|---|---|
| | 3. equipped green | 10.7 | * | 1. historical green | 10.5 | | 5. urban forestry | | 13,1 |
| | 2. big urban parks | 10.1 | | 4. decorative green | 7.5 | | 10. wood areas | | 13.2 |
| | 9. outdoor sports areas | 7.3 | | 12. cemeteries | 4,5 | | 11. uncultivated green suburban park, geological and botanical emergencies) | | 18.4 |
| | 6. urban services (health and education facilities, directional public green areas) | 4.0 | | 8. urban allottments | 0.7 | | | | |
| tot. | | 32.0 | | | 23.2 | | | | 44.7 |

*calculated on the total municipal green (public green areas+ natural protected areas, net to overlapping)

**Figure 4.** Map of current greenspaces in Ancona municipal area. Percentage composition of public green typologies and classification in three levels according the two qualitative indicators selected. Source: author's own elaboration from data taken by the municipality of Ancona, year 2018.

With reference to this categorization, the university campus areas are classified as 'urban services' (education facilities), and assigned to level 1 of usability.

At local scale, with reference to the experimental study conducted in the green areas of the university campus, from 2006, the type 1 areas gradually converted to type 2 and 3 (Figure 5), reducing maintenance routine operations. This was possible thanks to the application of the first principle of maintenance and diversified management.

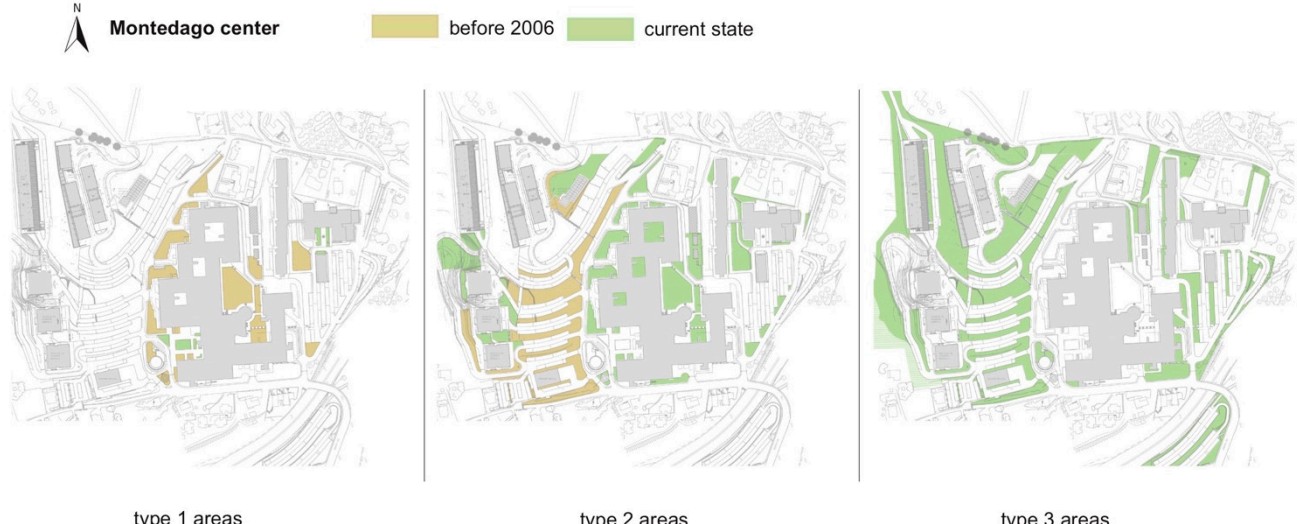

type 1 areas                     type 2 areas                     type 3 areas

**Figure 5.** Maintenance of turfgrass lawns: comparison between the three types of areas before and after the application of the diversified maintenance.

Diversified maintenance of turf lawn decreased by 23% in type 1 areas, while it increased by 36% type 2 areas and 80% type 3 areas (Table 6).

**Table 6.** Montedago center. Comparison between green areas divided by management categories (type 1, 2, 3) before 2006 and current state.

| University Campus Faculties | Type 1 Areas (m$^2$) | | Type 2 Areas (m$^2$) | | Type 3 Areas (m$^2$) | |
|---|---|---|---|---|---|---|
| | before 2006 | Current State | before 2006 | Current State | before 2006 | Current State |
| Montedago | 7640 | 710 | 13,363 | 14,920 | / | 40,570 |
| Torrette | 770 | 1738 | 230 | 3590 | 700 | 2883 |
| Centro, Villarey | 2992 | 3420 | 1016 | 1016 | 614 | 700 |
| Posatora | 6428 | 7706 | 1627 | 5972 | 8996 | 12,505 |
| Total | 17,830 | 13,574 | 16,236 | 25,498 | 10,310 | 56,658 |

For type 1 areas, mowing turns have been reduced from 25 to 15 per year. Considering that a turn grass mowing on the total green campus areas employs 43 man hours, we estimate a cost reduction from 1075 to 645 man hours.

For type 2 areas, mowing has been reduced from 15 to 7 times per year. Since a single turn requires the employment of 70 man hours, staff costs decreased from 1050 to 490 man hours.

Compared to the year 2006, the total cost reduction in annual routine maintenance activities for lawns corresponds to 990 man hours, equalling A decrease of 46% of the man hours. As one working day corresponds to 7 man hours, the total employment reduction is estimated to be 140 working days. As in our case a working day costs EUR 120.00, we estimate EUR 16,800 of total savings in lawn management. This is equal to approximately 16% of the total maintenance cost of green areas (to date, total investment in green management counts for approximately 860 working days per year).

From 2006 to 2019, total investments in machinery and equipment allowed a staff reduction from 9 to 6 workers and a decrease by 33% of the man hours.

The mechanical weed control based on a seasonal plan that followed the biological life cycle completely replaced herbicide applications in type 2 and 3 green areas, and the use of presowing weeding and of broad leaved species in type 1 areas. For cleaning streets and parking areas, the use of brushing-collecting machines once per month allowed the reduction in workers' employment from 1260 man hours to 350 man hours, equivalent to 130 working days and a financial saving of about EUR 15,000. In addition, the use of a manual brush cutter was reduced from four interventions to a single intervention per year, decreasing by 315 man hour and a cost saving of about EUR 5.400. It allowed the complete elimination of the use of chemical products and herbicides.

For type 1 areas, the disposal of 30 t of organic waste deriving from 15 mowings per year, costs about EUR 4500 per year.

Progressively, mulch mowing and mulching the leaves directly into the turf costs about EUR 500 per year, saving about EUR 4000 per year.

Chippers allowed the reusing of organic waste/residue from pruning as soil fertilizer and mulch, saving EUR 1000 per year and avoiding the purchase of four pallets of pine bark per year.

Considering the disposal costs of pruning waste of 12 cent plus VAT/q, in situ shredding permitted the reuse of an average of about 100 t of organic waste per year. We estimated cost savings per year of EUR 12,000 plus transport costs (mean of 100 times per year).

Total money-saving costs are about EUR 55,000 per year (considering the cost of EUR 82,142 in 2006 and EUR 27,172 in 2019), thanks to the application of first and second principles (Table 7).

**Table 7.** Cost benefits from sustainable management (2006–2019 period).

| Operation | Cost Reduction (Man Hours) | | Cost Reduction (%) | Money Saving Year 2019 (EUR) |
|---|---|---|---|---|
| | **Year 2006** | **Year 2019** | | |
| Turf management | | | | |
| Type 1 area | 1075 | 645 | 40 | 7370 |
| Type 2 area | 1050 | 490 | 53 | 9600 |
| Mechanical weed control | | | | |
| Brushing-collecting machines | 1260 | 350 | 72 | 15,600 |
| Use of manual brushcutter | 415 | 100 | 75 | 5400 |
| Mulch and leaf mowing | | | | 4000 |
| In situ shredding of pruning waste | | | | 12,000 + 1000 |
| Total saving (EUR) | | | | 54,970 |

Irrigation water consumption has also been greatly reduced. Irrigated lawns decreased from 6121 m² to 567 m² because of the degrading of type 1 areas to type 2 and 3.

Considering the 5 L/m² of medium daily water needed (for the spring/summer seasons, from May to September), water consumption decreased from about 3672 m³/year to about 340 m³/year, about 90% less water withdrawal, from 2006.

In addition, we avoided emergency irrigation systems in lawns, maintaining automated irrigation system only in type 1 areas (Figure 6).

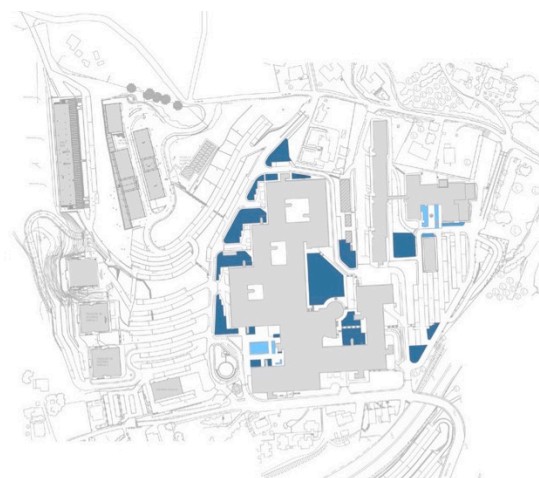

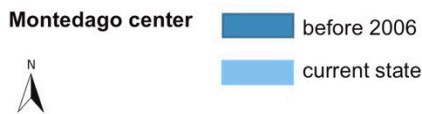

| City location | type 1 areas (m²) | | type 2 areas (m²) | |
|---|---|---|---|---|
| | before 2006 (m²) | current state (m²) | before 2006 (m²) | current state (m²) |
| Montedago | 630 | 500 | 5560 | - |
| Torrette | 1775 | 1775 | 935 | - |
| City center | 3545 | 3545 | 1016 | - |
| Posatora | 6142 | 6142 | 435 | - |
| Total | 12,092 | 11,962 | 7946 | - |

**Figure 6.** Sustainable water use: comparison between irrigated lawns before 2006 and after the application of the diversified maintenance principle (current state). The diagram on the left side refers to Montedago center.

## 4. Discussion

Data from the quantitative assessment method suggested a negative trend in urban green growth for the city of Ancona, with no increase in UGS from 2008. This critical situation of stasis largely contradicts the sustainable development goals established in the 2030 Agenda of the United Nations (2015) and the Urban Agenda for the European Commission, where the percentage of green areas is considered the first (and most important) indicator of sustainability in urban contexts.

Despite this, public green spaces required by national law and recognized as 'standards' in the land-use plan account for approximately 68% of the total public use areas (10.37 km² in 2014), highlighting the potential development of new green spaces according to land-use plan prescriptions.

For this reason, to think about new strategies to diversify maintenance and management, starting from the critical classification of green spaces by typology, level of usability, and level of management, will permit the better allocation of financial resources.

Therefore, the lowering of operating costs would allow an increase in the usability of the most strategic urban green areas; a necessary action, especially during the pandemic situation where the positive implication of the ES on human health and human wellbeing is particularly relevant [39].

According to this approach, the case study conducted in the green areas of the university campus shows a positive outcome, with important economic and environmental considerations on a city scale.

In fact, considering that the grasslands are the main component in terms of the size of the horizontal green, which represents 60% of the total green area of the university campus, the decrease by up to 65% of the total investment is significant. At the city scale, the total green spaces measure 29.2% of the municipal areas, equal to approximately 36 km². The areas classified as level 1 (according to our method) represent about 32%, equal to 36.45 km².

Here, the green areas of the university campus that belong to this category represent approximately 1% (approximately 0.11 km²). Therefore, regarding the case study analyzed, since the long term application of thirteen years made it possible to record a reduction in the costs of the management of green areas equal to EUR 55,000, on city scale a potential decrease in the costs of about EUR 5,000,000 in ten years could be estimated, equal to about EUR 400,000 per year in urban green management.

Furthermore, as demonstrated for the local scale, the diversified maintenance approach has resulted in a very significant decrease in the total cost of lawn maintenance.

If this cost-saving allows for the management of approximately 40% more external areas on a local scale, the same potential percentage increase in public green spaces with similar characteristics could be expected.

In perspective, as the green spaces categorized in level 1 and level 2 (according to the usability and management indicators selected in this study) account for about 20 km$^2$, equal to about 55.2% of total public green surfaces and almost entirely managed by the municipality, the application of a diversified maintenance approach at the city scale could bring a relevant reduction in public financing.

In addition, the percentage of green areas categorized in levels 1 and 2 could represent the 'workspaces', where the optimization of management strategies, defined in the green management plan, will help to design the GI and support the implementation of green infrastructure.

The application of the diversified maintenance of grassland associated with a lower cutting regime benefits not only the economical sustainability of UGS but also the environment [40]. This has positively contributed to improving land cover and biodiversity at the city scale.

In fact, from 2006, in type 2 and type 3 areas, avoiding the intensive mowing required by monoculture [41] led to a cost saving of about half of the previous costs, but also provided the formation of permanent multiphase lawns, with the increase in the various types of floral species and aided in weed control [42]. Furthermore, as turfgrass lawns comprise 70–75% of UGS worldwide [43], the development of various alternatives that avoid intensive turfgrass management could be very important to understand the beneficial effect on population dynamics, the community structure, and biodiversity [41,44].

The natural evolution from grass lawn to flower meadow during the year (from type 1 areas to type 2 and 3 areas) led to the reduction in grasses in favor of leaf species such as *Trifolium repens* and flowering species such as *Taraxacum officinale*, *Bellis perennis*, *Ranunculus ficaria*, *Hedysarum coronarium* L. for type 2 areas (Figure 7a,b). This species increase was triggered by the reduction in cutting frequency (management regime of four cuttings per year) provided in 2019 [45].

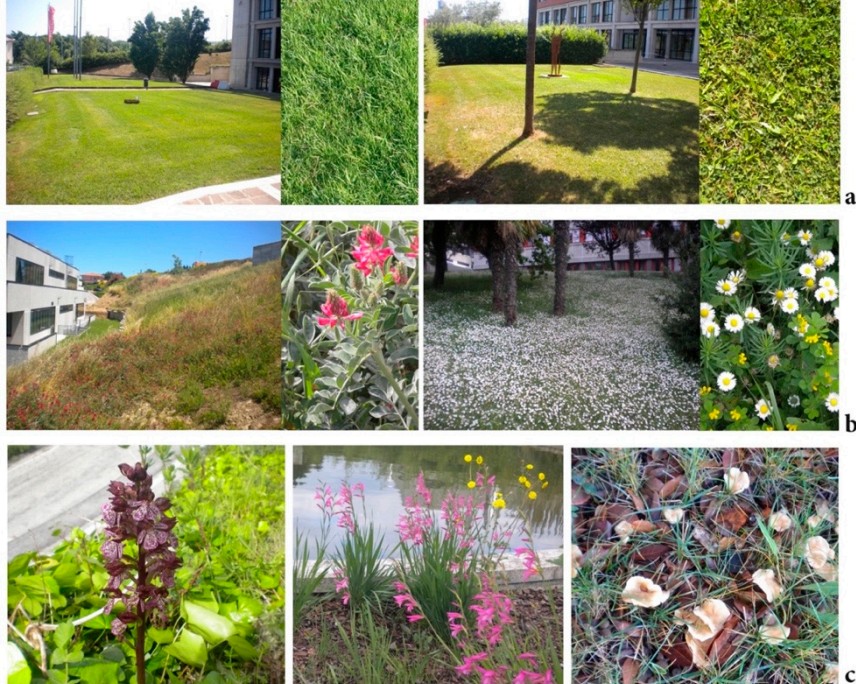

**Figure 7.** (**a**) Comparison between 2006 and current state of lawn of Faculty of Medicine. (**b**) Montedago center. Evolution of natural meadow: green areas type 3 (left side) and type 2 (right side). (**c**) New spontaneous species: *Gladiolus italicus*, *Orchis purpurea* and mushrooms (from left to right).

Furthermore, since 2016, the progressive reduction in the periodic mowing of the grass according to the biological life cycle has allowed the complete elimination of herbicides and led to the growth in spontaneous species such as *Gladiolus italicus*, *Himatoglossum robertianus*, *Orchis purpurea* and mushrooms (Figure 7c).

IN addition, as the need for water is growing rapidly and water scarcity is a serious problem in major parts of the world [46], the avoidance of an irrigation system in the lawn constitutes a significant contribution to the public water demand, which may experience an estimated 40% supply shortage by 2030 [47].

Replacing intensive management with alternative turf management regimes could offer greater habitat opportunities for both plants and insects [48] and provide clear benefits to pollinator and wild bee populations [49]. This type of management can also be compatible with the use of different living mulches, which is potentially productive, as was recently found in organic fruit orchards [50,51].

From 2006, the application of alternative solutions that completely replaced herbicide applications in type 2 and 3 green areas also occurred, to improve soil fertility.

In fact, the progressive reduction in mechanical maintenance operations has contributed to reducing fertilization and to improving the chemical, physical and biological structure of the soil [52], as well as being effective in terms of cost and time. Furthermore, reusing mulch mowing and leaf mulching as organic material directly into the turf increased soil organic matter, and helped improve the aesthetics of the turf [53,54] and the benefits for urban biodiversity (Figure 8a,b).

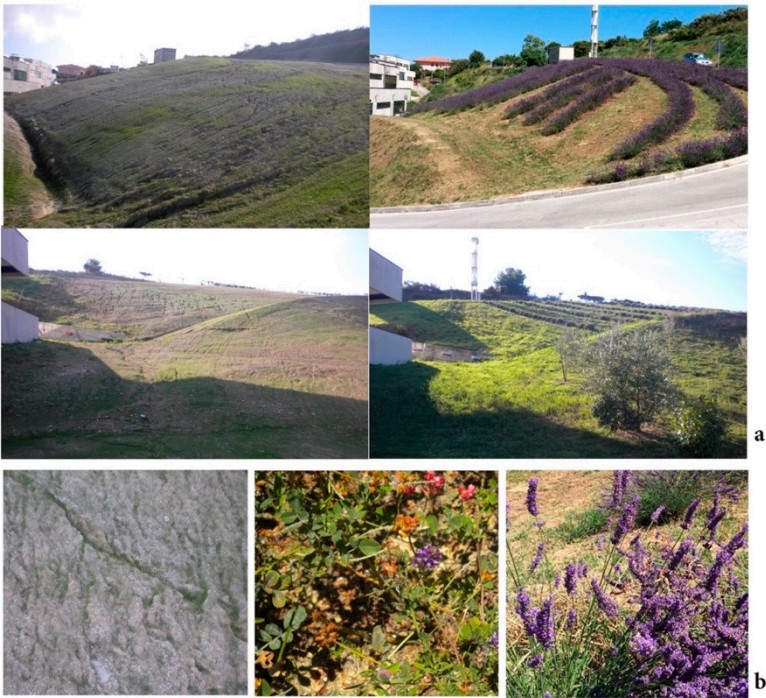

**Figure 8.** (**a**) Evolution of soil cover: comparison between 2013 (left) and 2018 (right). (**b**) Detail of soil (from left to right): starting condition (year 2010); spontaneous evolution of mowing after the hydroseeding (year 2013); newly cultivated areas of a lavender field. Green manure with spontaneous vegetation was applied between the row (year 2019).

In type 3 areas, low intensity management benefits the richness of plant species in urban grasslands [55,56].

The planned shredding geared to preserving the complexity of the vegetation structure and selected native shrubs helped support multiple taxa and promotes abundant and diverse assemblages of insects and birds [57]. Soil acts as a natural source and sink for greenhouse gases (GHG), which are responsible for global warming and climate change.

As green areas and the agricultural sector have an important impact on GHG (e.g., $CO_2$, $CH_4$, $N_2O$) emissions, the definition of mitigation strategies is needed, especially for the Mediterranean climate areas that appear most vulnerable to climate change. The introduction of perennial legumes, such as alfalfa (*Medicago sativa* L.), falls within this scope [58].

In this case, the management interventions aimed at preserving native plants have contributed to: (a) controlling the vegetative cycle, which has improved the stability of the soil, in particular in the case of slopes (Figure 9a); (b) supporting a greater diversity of insects and, therefore, insectivorous birds [59], which enhances biodiversity and the local landscape (Figure 9b). On the contrary, 'landscaping' oriented to the replacement of native plants with non-native ornamental plants has the ability to disrupt urban food webs [60].

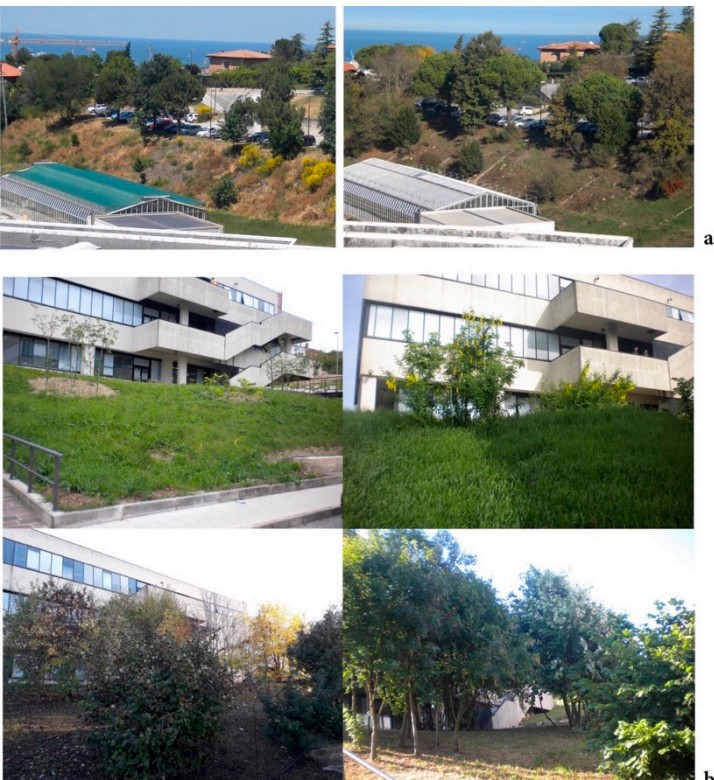

**Figure 9.** (**a**) Montedago center. Evolution of slopes: comparison between year 2013 (left) and year 2018 (right); (**b**) example of an urban wood management, with planting of native species such as linden, maycindol, hackberry, ash, elm. From right (up) to left (down): comparison between year 2010, 2013, 2018 (fall), 2019 (spring).

At the urban scale, the natural transition from the meadow heavy landscape to more natural looking landscapes based on wildflower meadows can offer more ecological benefits and improve urban biodiversity, despite also altering the appearance and usability of the respective greenspaces.

This attitude must have a positive public response and acceptance by the residents. In fact, the value that a 'city user' attributes to each type of urban lawn based on personal preferences strongly conditions management regimes [61].

Tall grass urban lawns can support considerably higher levels of native biodiversity than short lawns [62–64], but they require human acceptance of a 'more disordered' aesthetic concept of green areas. Therefore, the agreement with converting short cut lawns into biodiversity friendly tall grass meadows largely depends on the ecological and sociocultural values that the residents assign to different types of urban grasslands, as well as the ways in which they use greenspaces [61].

As reported in the Italian Ministerial Decree for public green spaces, it is necessary to encourage dialogue and participation in management strategies and decision-making processes. This 'bottom up' strategic action contributes to awareness rising, human acceptance and will strengthen the relationship between managers and users [65].

To facilitate this, to take care of some relevant detail in design and management, such as mowing mown borders and mown paths transecting a meadow, could increase the acceptance of this type of urban grassland [66].

Whereas the values that people assign to urban nature are important when planning the conservation of biodiversity in public green spaces [57,67–69], this contribution can be helpful in raising awareness of alternative UGS management practices that need the positive relationship of wilderness and species richness with tall grass lawns and social and nature related green space activities.

This would represent a necessary transition to enhance benefit for urban biodiversity and lead to richer wildlife communities [70].

## 5. Conclusions

The case study shows that thirteen years of the application of alternative management strategies in green areas can bring economic and environmental benefits at the urban scale.

In particular, the diversified maintenance of the grass lawns has made it possible to reduce the costs that the expanded green areas managed, according to different levels of usability.

Therefore, adopting a *multi scale* approach [71], from the local scale of experimentation to the urban dimension of green infrastructures, would allow the transference of all the benefits deriving from the ecological and biodiversity process to urban contexts, preserving a balance with economic and human needs.

Since UGS must be planned and managed within a common theme as part of a long term green space strategy for the whole city [72], adopting a coordinated management plan that expands a network of high quality UGS would have a great impact on cost reduction and is essential to preserving biodiversity, which must be managed on multiple scales.

To transfer the diversified maintenance approach to urban green spaces, starting from diversification in terms of the level of usability and management can lead to a 'widespread' sustainability of green spaces at the urban scale [73] (Figure 10). In addition, the application of new management activities across different UGS can ultimately result in conversion to habitats that require less annual maintenance and cost [74].

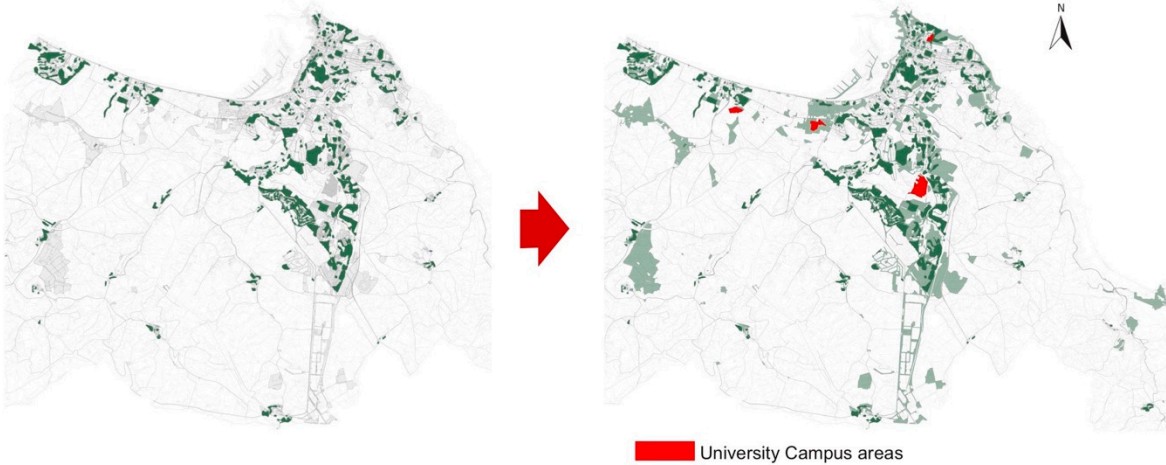

**Figure 10.** Existing UGS in Ancona and potential net of UGS increased with green areas of university (red) and other public green areas. Source: author's elaboration on data from the Municipality of Ancona, year 2018.

For this reason, the role of public institutions such as universities can be essential for coordinating strategies on both public and private parts of the green infrastructure. This will also reduce the high level of fragmentation of the urban greenery, which is typical of urban contexts.

Hence, identifying a unified management strategy to be applied in continuous areas of the city can help to achieve better results in support of sustainability, and to redesign the continuity of GI as a core principle of its planning [75].

Under such an approach, the case study can be replicable for other similar urban contexts. Extending the principles and best practices to major regional cities will lead to positive growth in the quality of urban life for the most resilient cities. It will contribute to develop strategies for future green master plans, in line with the need to enhance the 'sustainability performance' of green spaces by acting in local contexts and, in particular, urban contexts [76–80].

It is also highlighted that there is a need to introduce a strategic vision of a green system within the land-use plan, which is the current tool for the management of urban and peri-urban areas on a municipal scale [81].

**Author Contributions:** Conceptualization, M.P., G.M., A.M., R.S. and D.N.; methodology, M.P., G.M., A.M., R.S. and D.N.; validation, G.M., A.M. and D.N.; data curation, M.P.; writing—original draft preparation, M.P.; writing—review and editing, M.P.; visualization, M.P.; supervision, G.M., A.M. and D.N. All authors have read and agreed to the published version of the manuscript.

**Funding:** This research received no external funding.

**Institutional Review Board Statement:** Not applicable.

**Informed Consent Statement:** Not applicable.

**Data Availability Statement:** Raw data used in this study have been sourced from open data platform, as indicated in the article. Processed data are available on request from the corresponding author.

**Acknowledgments:** The development of the experimental study was managed by the team of experts and technicians of "Experimental farm P. Rosati", Polytechnic University of Marche, Ancona, Italy. We thank all the workers and farm technicians Giuseppe Siciliano (responsible for campus green infrastructures and agronomist) and Giorgio Murri (farm coordinator and agronomist) for the monitoring of the areas during the period of experimentation. The comparative photos belong to Giuseppe Siciliano.

**Conflicts of Interest:** The authors declare no conflict of interest.

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
