# Peer review of "Sustainable Management Practices for Urban Green Spaces to Support Green Infrastructure: An Italian Case Study"

_sustainability, doi:10.3390/su14074243_

Round 1
Reviewer 1 Report
The paper has a much promising title – it merges the topic of sustainable management practices for Urban Green Spaces with the support of Green Infrastructure. In fact, it is a case study on the reduced costs and roughly described environmental benefits of reduced mowing of the lawns at the university campus location in Ancona. The topic is accurate and refers to the actual challenge that many cities and places are facing, but is not very original – it is widely accepted now that less maintenance means more environmental benefits and fewer costs. And that we need biodiversity at many scales of UGS.
Authors, in general, used the proper structure of the paper and referred to accurate literature (which however could be improved by the references directly linked with the topic of the case study, that is to say: lawn maintenance in the context of UGS management in the city scale).
The case study is interesting and provides some relevant data. It should be illustrated also however in graphic form (tables presenting expenses works carried would make the presentation more clear) and referring to some further quantitative measures.
The paper needs definitely some English editing since there are some sentences that are unclear. Also, the logic of the text has to be improved – for example, the whole introduction is composed of scattered paragraphs, which although related to the topic and one with the other, form a very long text that does not convince the reader that the research presented fills the knowledge gap.
Authors at first make an introduction in which they link the issues of Green Infrastructure and Urban Green Spaces with the wide topic of sustainability and the conflicts between economic growth and environmental protection. They also stress the importance of green Infrastructure from the social point of view to finally conclude that “most significative studies at non-EU, EU and national level evaluate sustainability of UGS and therefore of GI by a set of specific indicators mainly based on a ‘quantitative evaluation approach’.”
Authors suggest that they will use “management indicators to evaluate sustainability of UGS” . Furthermore, they state correctly, that “There is lack in the linkage between economic result and best management practices that can bring positive implication in ecological field and urban biodiversity conservation”. Authors claim that
aims to: a) discuss a theoretical model based on a list of good practices to manage UGS in a more sustainable way; there is no theoretical model is discussed – it is a case study
- b) demonstrate the replicability of the model at the scale of GI; there is no replicability demonstrate – there is a possibility shown
- b) provide guidelines to build management indicators that measure the sustainability of urban green space, which implement the set of urban quality indicators already present in the literature.
[typo: Should be c)] no guidelines are provided – there is just a statement at the end, that the guidelines are needed. There is no suggestion on how to link the management practices with quantitative indicators mentioned at the beginning of the study. Also, there is no comment on how the result of the case study would refer to the issues of usability and accessibility – while there are some studies on social perception and peoples’ habits concerning less maintained green areas.
It is a generally interesting issue – that Authors referring to the types of UGS in Ancona categorized them regarding “usability” – and in the group of less “usable” UGS, they put the most natural and most vegetated ones (and surprisingly… beaches as well). It is also worth mentioning that as less “usable” green spaces are classified here also those which obviously need less maintenance. And the conclusion of the paper (which I do not question!) is that less maintenance is good for many reasons. Maybe the types of UGS should not be labeled according to the “usability” category? Or maybe that should not be the dominant category?
Other detailed remarks:
LL 131- 133 sentence not clear - you can present/describe a method "this paper discusses a method to demonstrate that improving sustainability of UGS represent a necessary step to improve sustainability of GI and ‘green performance’ at urban scale"
LL 139 - 142 sentence unclear "To evaluate the sustainability of UGS trough the definition and development..."
L 182 "the highest percentage of land use" and further in L186 "land consumption" - do you mean urbanization and/or increasing the building area?
L 222 - "municipal surface" do you mean surface of the city or areas in the ownership of the city (public property)?
Figure 1. colors of "other urban services" and "urban forestry and historical green and decorative green areas are almost the same - one cannot distinguish them on the map.
Also the categories are not clear - Urban parks are in Level 1 while Historical green is in Level 2 - I suppose that there are some parks in Ancona which are historic parks - to which category they would belong?
Figure 2 - did you mean "public GREEN areas in the city Ancona"?
L 261 - shouldn't be that "Step 2"?
L 292 - ". Data were taken from 2006 to 2019 in the University campus" - can you show the data collected in any tables? It would well organize the presentation of the case study
L 460 "Observing the comparison between the pictures taken..." were there any initial observations or investigations carried out regarding biodiversity of the lawns? or it is just the comparison of photographs - which is far insufficient to draw scientific conclusions without any methodological explanation...
L 50 9 i 510 (sx did jou mean left?) (dx) right?
Figuire 9 - this should be entitled - "existing UGS in Ancona and potential net of UGS inreased with green areas of university (red) and other public institutions - please provide another shade of green for that...
worth reading:
J. Lampinen et al., "Acceptance of near-natural greenspace management relates to ecological and socio-cultural assigned values among European urbanites", Basic and Applied Ecology 50 (2021) 119-131
Mollashahi H, Szymura M, Szymura TH(2020) Connectivity assessment and prioritizationof urban grasslands as a helpful tool for effectivemanagement of urban ecosystem services. PLoSONE 15(12): e0244452. https://doi.org/10.1371/journal.pone.0244452
Lonati, M., Probo, M., Gorlier, A., Pittarello, M., Scariot, V., Lombardi, G. and Ravetto Enri, S. (2018). Plant diversity and grassland naturalness of differently managed urban areas of Torino (NW Italy). Acta Hortic. 1215, 247-254
DOI: 10.17660/ActaHortic.2018.1215.44
https://doi.org/10.17660/ActaHortic.2018.1215.44
Melissa Sehrt, Oliver Bossdorf, Martin Freitag, Anna Bucharova,
Less is more! Rapid increase in plant species richness after reduced mowing in urban grasslands, Basic and Applied Ecology, Volume 42, 2020, Pages 47-53, ISSN 1439-1791, https://doi.org/10.1016/j.baae.2019.10.008.
(https://www.sciencedirect.com/science/article/pii/S1439179119302932)
Palliwoda, J., and J. A. Priess. 2021. What do people value in urban green? Linking characteristics of urban green spaces to users’ perceptions of nature benefits, disturbances, and disservices. Ecology and Society 26(1):28.
https://doi.org/10.5751/ES-12204-260128
Biernacka, Magdalena and Kronenberg, Jakub (2019) "Urban Green Space Availability, Accessibility and
Attractiveness, and the Delivery of Ecosystem Services," Cities and the Environment (CATE): Vol. 12: Iss. 1,
Article 5.
Available at: https://digitalcommons.lmu.edu/cate/vol12/iss1/5
Fengping Yang, Rethinking lawns as prevalent elements of urban green spaces, Doctoral thesis, Swedish University of Agricultural Sciences, Uppsala 2019, Acta Universitatis agriculturae Sueciae, 2019:6, ISSN 1652-6880 , online: https://pub.epsilon.slu.se/15834/1/yang_f_190114.pdf
Reviewer 2 Report
THE PAPER PRESENTS MANY INTERESTING POINTS. IT IS ADVISABLE TO REVIEW IT AS A WHOLE WITH A SCIENTIFIC APPROACH DEFINING THE CONTRIBUTION TO RESEARCH
Reviewer 3 Report
This article appears to be a case study of costs of sustainable management for urban green spaces. The case study seemed to be of various green areas on the university campus. However, the main aims of the case study were not clear until the Discussion.
The authors should substantially reorganize and edit the Background and Methods sections to clarify the aim of the case study.
On page 2. line 93, the authors discuss evaluating sustainability. They should consider discussing their study at this point. The excess background makes it difficult to find the relevant details.
On page 3 the authors discuss the aims of the paper. The authors should make one of these aims directly related to the case study itself. Additionally, part b did not appear to be addressed.
There does not appear to be a "step 2" between pages 5 and 7.
Did the authors assign "usability" themselves based on the criteria?
This may not be possible, but the authors should carefully consider the figures included as they are not completely clear when printed in grayscale.
Reviewer 4 Report
The text is organised in chapters which are roughly balanced and punctuated by some illustrations. In this respect, the whole is satisfactory.
Round 2
Reviewer 1 Report
Authors significanltly improved their article - congratulations!
A completely rewritten introduction is now coherent and introduces very well the topic of the paper and relevant background. It situates clearly described research within the broader context.
Also improved section on materials and methods provides the reader with a good explanation of tools and methods used by authors to carry out and analyse their study.
A newly written part of the discussion in particularily valuable since it shows results of the study in a wider GI planning instruments and urban planning regulations.
All in all the paper gained a new quality and is definitely worth publishing as an important voice on management of green spaces and its implications at urban scale in context of the world's growing urbanisation and sustainability issues with that fact.
Author Response
The introduction has been re-edited and has been more focused on Italian land-use planning instruments derived from the Italian Ministerial Decree. Furthermore, the first part of the introduction has been shortened and better quoted.
Also, the discussion has been specified, in particular, text from L 430-450, and from L 551-576.
Abstract has been changed.
Reviewer 2 Report
The paper after revision, has been much improved and clarifies the research question. It is advisable to update the bibliography also in relation to many papers recently published on Sustainability and Climate having as keywords green infrastrure, tree, multicriteria in order to complete the updated literature.
Author Response

(The authors gave the same response as above.)

Reviewer 3 Report
The authors revisions greatly improved the clarity from the last version. Below are some additional comments:
- The introduction is now more organized and the motivation seems to be that "management of urban lawns is the main component of UGS" and that green census, green Masterplan, and green regulation are meant to fill the gap between maintenance and strategic planning. However, the authors should still substantially edit down the Introduction to be more Italy focused as the instruments are from the Italian Ministerial Decree. Some knowledge of sustainability history and and goals can be assumed in this journal and would focus the goals of the study.
- In the Materials and Methods section, everything from p 3, line 138-160, 187-191 do not seem like Methods. They appear more as background or Results and should be moved to the appropriate section, relabeled, or removed if it is extraneous to the goals of this study.
- The beginning of the Discussion seems to contradict itself. The authors claim there is a "negative trend" on line 473 and then in the following paragraph notes what is required and states that there is "a potential growth of green infrastructures." Authors should clarify these statements.
- On line 598, the authors state that this needs to have acceptance from the residents and than state the residents find it "untidier." Authors should discuss the implications for that or if that means that they don't agree with the possible sustainability practices.
Round 3
Reviewer 3 Report
The author edits substantially improved the clarity of the manuscript. Minor comments are below:
I. In the abstract, please state the words before using the abbreviations like GI. They are spelled out in the body of the paper but not the abstract.
2. Line 66: Authors should define "this instrument." I think it refers to the quanititative instrument but it is not clear.
3. Line 95: Authors may want to insert the words "may be" before effective as the next sentence states that there is a need to evaluate effectiveness.
